# Solving Schrödinger Bridges via Maximum Likelihood

**DOI:** 10.3390/e23091134

**Published:** 2021-08-31

**Authors:** Francisco Vargas, Pierre Thodoroff, Austen Lamacraft, Neil Lawrence

**Affiliations:** 1The Computer Laboratory, Department of Computer Science and Technology, University of Cambridge, William Gates Building, 15 JJ Thomson Avenue, Cambridge CB3 0FD, UK; pt440@cam.ac.uk; 2The Cavendish Laboratory, Deparment of Physics, The Old Schools, Trinity Ln, Cambridge CB2 1TN, UK; al200@cam.ac.uk

**Keywords:** Schrödinger bridges, machine learning, stochastic control

## Abstract

The Schrödinger bridge problem (SBP) finds the most likely stochastic evolution between two probability distributions given a prior stochastic evolution. As well as applications in the natural sciences, problems of this kind have important applications in machine learning such as dataset alignment and hypothesis testing. Whilst the theory behind this problem is relatively mature, scalable numerical recipes to estimate the Schrödinger bridge remain an active area of research. Our main contribution is the proof of equivalence between solving the SBP and an autoregressive maximum likelihood estimation objective. This formulation circumvents many of the challenges of density estimation and enables direct application of successful machine learning techniques. We propose a numerical procedure to estimate SBPs using Gaussian process and demonstrate the practical usage of our approach in numerical simulations and experiments.

## 1. Introduction

Analysis of cross-sectional data is ubiquitous in machine learning and science. Temporal data are typically sampled at discrete intervals due to technological or physical constraints. This means information between time points is lost. This motivates the need to model the stochastic evolution of a process between sampled time points. The classical Schrödinger bridge problem [1,2] finds the most likely stochastic process that evolves a distribution $\pi_{0}\left(x\right)$ to another distribution $\pi_{1}\left(y\right)$ consistently with a pre-specified Brownian motion. We consider a more general dynamical Schrödinger bridge problem for *any* pre-specified diffusion prior. Practically, this generalization allows us to exploit domain knowledge, e.g., oceanic and atmospheric flows might be interpolated from empirical measurements using previously established dynamics as priors.

In the classical set up numerical approaches to solve the Schrödinger bridge are mainly based on the Sinkhorn–Knopp algorithm [3]; however, extending those algorithms to more general diffusion priors and marginals requires complex adaptations. We introduce an iterative proportional maximum likelihood (IPML) algorithm to solve the general Schrödinger bridge problem. The IPML algorithm also obtains a good approximation for the dynamics of the underlying physical process that solves the SBP. This contrasts to previous approaches [4,5,6] that estimate the value that extremises the SBP objective. Practically, this means we obtain physically interpretable solutions that we can leverage for downstream tasks.

IPML’s inspiration is from *probabilistic numerics* (PN) [7]. We combine a PN styled formulation with the iterative proportional fitting procedure (IPFP) [8,9]. The algorithm iteratively simulates trajectories that converge to the SBP. We prove that IPML converges in probability at each iteration. Our numerical experiments show the algorithm can be implemented with Gaussian process (GP) models of the drifts. GPs allow us to incorporate functional prior information. We demonstrate the practical use of our algorithm on real-world embryoid cells data (see Figure 4) by quantitatively and qualitatively comparing the performance of our algorithm to state-of-the-art deep learning methods and optimal transport techniques. To summarise, the main contributions of our work are:We recast the iterations of the dynamic IPFP algorithm as a regression-based maximum likelihood objective. This is formalised in Theorem 1 and Observation 1. This differs to prior approaches such as [10] where their maximum likelihood formulation solves a density estimation problem. This allows the application of many regression methods from machine learning that scale well to high dimensional problems. Note that this formulation can be parametrised with any method of choice. We chose GPs; however, neural networks would also be well suited,We solve the aforementioned regression objectives using GPs [11] motivated by the connection between the drift of stochastic differential equations and GPs [12],We provide a conceptual comparison with the approach by [10] and detail why the density estimation formulation in [10] scales poorly with dimension,Finally we re-implement the approach by [10] and compare approaches across a series of numerical experiments. Furthermore we empirically show how our approach works well in dimensions higher than 2.

## 2. Technical Background

Our solution has three components. (1) We reformulate the SBP as a dynamical system giving a stochastic differential equation (SDE) with initial value (IV) and final value (FV) constraints [13]. (2) We reverse the system, reformulating the FV constraint as an IV constraint. Both IV problems are solved through a stochastic control formulation (Section 2.1.2). (3) We iterate the IV and FV constrained problems to converge to the full boundary value constrained SDEs.

### 2.1. Dynamic Formulation

The dynamic version of the Schrödinger bridge is written in terms of measures over the space of trajectories, which describe the stochastic dynamics defined over the unit interval.

**Definition** **1.**
*(Dynamic Schrödinger problem) The dynamic Schrödinger problem is given by*

(1)
$$\begin{array}{c} \begin{array}{c} \underset{Q\in D(\pi_{0},\pi_{1})}{\inf}D_{\mathrm{KL}}\left(Q||Q_{0}^{\gamma }\right), \end{array} \end{array}$$

*where $Q\;\;\in \;\;D(\pi_{0},\pi_{1})$ is a measure with prescribed marginals of $\pi_{0},\pi_{1}$ at times 0,1 that is ${\left(X_{0}\right)}_{\#}Q=\pi_{0}$ and ${\left(X_{1}\right)}_{\#}Q=\pi_{1}$. $Q_{0}^{\gamma }$ is a drift augmented Brownian motion with a scalar volatility γ acting as the prior, see Figure 1. Traditionally $Q_{0}^{\gamma }\;:=\;\;W^{\gamma }$ is a Wiener measure with volatility γ, however we consider the general setting in this work.*


The prior $Q_{0}^{\gamma }$ can be written as a solution of the diffusion: (2)$$\begin{array}{c} \begin{array}{c} dx\left(t\right)=b_{0}^{+}\left(x\left(t\right),t\right)dt+\sqrt{\gamma }d\beta^{+}\left(t\right),\;\;\;x\left(0\right)∼\pi_{0}^{Q_{0}^{\gamma }}. \end{array} \end{array}$$
From here we use $b_{0}$ to denote the drift of the prior $Q_{0}^{\gamma }$. Note that a finite KL implies Q and $Q_{0}^{\gamma }$ are both Itô SDEs with volatility γ. The diffusion coefficient γ is a time homogeneous constant and for the KL-divergence to be finite the process Q must have γ as its diffusion coefficient.

#### 2.1.1. Time Reversal of Diffusions

When sampling solutions, IV constraints are trivially solved by initialisation of samples, but FV constraints are more problematic; however, if we can reformulate the SBP as a *time reversed diffusion* the FV constraint becomes an IV constraint (Section 3.2). Here we review how the diffusion is reversed. We reverse time (see Figure 1) in the random variable x(t) described by the diffusion in Equation (Equation 2) such that $x^{-}\left(t\right)=x\left(1-t\right)$. The reverse time diffusion $x^{-}\left(t\right)$ is also an Itô process. For a modern application of time reversal in machine learning see [14].

**Lemma** **1**([15])**.**
*If $x^{+}\left(t\right)$ obeys the SDE*
$$\begin{array}{c} \begin{array}{c} dx^{+}\left(t\right)=u^{+}\left(x^{+}\left(t\right),t\right)dt+\sqrt{\gamma }d\beta^{+}\left(t\right),\;\;\;x^{+}\left(0\right)∼\pi_{0}^{Q_{0}^{\gamma }}, \end{array} \end{array}$$
*then $x^{-}\left(t\right)=x^{+}\left(1-t\right)$ obeys*
$$\begin{array}{c} \begin{array}{c} dx^{-}\left(t\right)=u^{-}\left(x^{-}\left(t\right),t\right)dt+\sqrt{\gamma }d\beta^{-}\left(t\right),\;\;\;x^{-}\left(0\right)=x^{+}\left(1\right). \end{array} \end{array}$$
*where $\beta^{-}\left(t\right)$ is a Brownian motion adapted to the reverse filtration ${\left(F_{i}^{-}\right)}_{i\in T}$, that is $F_{t}^{-}\;\;\subseteq F_{s}^{-},\;s\;\leq \;t$. Furthermore, the dual drift $u^{-}\left(x,t\right)$ satisfies Nelson’s duality relation:*
(3)$$\begin{array}{c} \begin{array}{c} u^{+}\left(x,t\right)-u^{-}\left(x,t\right)=\gamma \nabla_{x}\ln p\left(x,t\right), \end{array} \end{array}$$
*where p(x,t) solves the associated Fokker–Planck equation.*

**Proof.** A variety of proofs can be found for this result [15,16,17,18,19]. Note the formulation of Equation (Equation 3) varies slightly across studies we use the one from [15,16] (see [15] (p. 87), definition of osmotic velocity). □

#### 2.1.2. Stochastic Control Formulation

Now that the FV constraint has been converted to an IV constraint, we cast the problem into a stochastic control formulation to estimate the drift of each diffusion process. Following from the dynamic formulation, the control formulation casts the problem explicitly in terms of stochastic differential equations. The control formulation is used to enforce constraints as initial value problems. Furthermore, the drift based formulations of the SBP admit a reverse time formulation, which starts the chain at the end of the interval and progresses the dynamics backwards in time to the start.

**Lemma** **2**([20])**.**
*Let the measure Q be defined by solutions to the SDE*
$$\begin{array}{c} \begin{array}{c} dx^{+}\left(t\right)=b^{+}\left(t\right)dt+\sqrt{\gamma }d\beta^{+}\left(t\right),\;\;\;x^{+}\left(0\right)∼\pi_{0}^{Q} \end{array} \end{array}$$
*Then the KL divergence $D_{\mathrm{KL}}\left(Q||Q_{0}^{\gamma }\right)$ can be decomposed in terms of either the forward or reversed diffusion as*
(4)$$\begin{array}{c} \begin{array}{c} D_{\mathrm{KL}}\left(Q||Q_{0}^{\gamma }\right)=D_{\mathrm{KL}}(\pi_{\frac{1\mp 1}{2}}^{Q}\left|\right|\pi_{\frac{1\mp 1}{2}}^{Q_{0}^{\gamma }})+E_{Q}\left[\int_{0}^{1}\frac{1}{2\gamma }||b^{\pm }\left(t\right)-b_{0}^{\pm }\left(x^{\pm }\left(t\right),t\right)|{|}^{2}dt\right], \end{array} \end{array}$$
*where $x^{-}\left(t\right)$ is the time reversal of $x^{+}\left(t\right)$, and $x^{+}\left(1\right)∼\pi_{1}^{Q}$.*

**Proof.** This theorem follows by a direct application of the disintegration theorem (Appendix B) followed by Girsanov’s theorem. A detailed proof can be found in [20]. □

Using the above decompositions, we can solve the SBP by minimising either decomposition in Equation (Equation 4) over the space of random processes $b^{\pm }\left(t\right)$ that satisfy a valid Itô SDE drift. The backwards and forwards objectives are respectively:(5)$$\begin{array}{c} \begin{array}{cc} \min_{Q\in D(\pi_{0},\pi_{1})}D_{\mathrm{KL}}\left(Q||Q_{0}^{\gamma }\right) & =\min_{b^{\pm }\in B}E_{Q}\left[\int_{0}^{1}\frac{1}{2\gamma }||b^{\pm }\left(t\right)-b_{0}^{\pm }\left(x^{\pm }\left(t\right),t\right)|{|}^{2}dt\right],\\ s.t.\;\;\;dx^{\pm }\left(t\right)=b^{\pm }\left(t\right)dt+\sqrt{\gamma }\beta^{\pm }\left(t\right), & \;\;x^{+}\;\left(0\right)\;∼\;\pi_{0},\;\;x^{+}\;\left(1\right)\;∼\;\pi_{1},\;\;x^{-}\;\left(1\right)\;∼\;\pi_{0},\;\;x^{-}\;\left(0\right)\;∼\;\pi_{1}. \end{array} \end{array}$$

While we do not directly use the stochastic control formulations, the drift-based formulation serves as inspiration for our iterative scheme. Specifically, the existence and parametrisation of an optimal drift as shown in [10,20].

**Lemma** **3**([20])**.**
*The objectives in Equation (Equation 5) have optimal drifts:*
(6)$$\begin{array}{c} \begin{array}{c} b^{+*}\left(t\right)=\gamma \nabla_{x}\phi \left(x^{+}\left(t\right),t\right),\;b^{-*}\left(t\right)=\gamma \nabla_{x}\hat{\phi }\left(x^{-}\left(t\right),t\right) \end{array} \end{array}$$
*where the potentials $\phi ,\hat{\phi }$ solve the Schrödinger system [20].*

This Lemma is key in formulating our ML approach to IPFP since it justifies our parametrisation of the drift in terms of a deterministic function $u^{\pm *}$ i.e., $b^{\pm *}\left(t\right)=u^{\pm *}\left(x^{\pm }\left(t\right),t\right)$. For a brief introduction to the Schrödinger system and potentials see Appendix A.

### 2.2. Iterative Proportional Fitting Procedure

We now have two boundary value constrained diffusion processes solved through a stochastic control formulation. We use the iterative proportional fitting procedure (IPFP) to alternate between the forward and backward formulation with only one initial value constraint enforced at a time (see Figure 2), such that convergence to the full boundary value problem is guaranteed.

The measure theoretic version of IPFP we introduce is an extension of the continuous IPFP, initially proposed by [8] to a more general setting over general probability measures (see [21,22]). The convergence of IPFP has been shown in [9] and further extensions and results have been presented in [21,22]. The idea behind this family of approaches is to alternate minimising KL between the two marginal distribution constraints,
$$\begin{array}{c} \begin{array}{c} P_{i}^{*}=\underset{P\in D(\cdot ,\pi_{1})}{arg inf}D_{\mathrm{KL}}\left(P|\right|Q_{i-1}^{*})\;Q_{i}^{*}=\underset{Q\in D(\pi_{0},\cdot )}{arg inf}D_{\mathrm{KL}}\left(Q|\right|P_{i}^{*}), \end{array} \end{array}$$
until convergence. The quantities $P_{i}^{*}$ and $Q_{i}^{*}$ in the algorithm of Figure 2 are known as half bridges [10] and can be expressed in closed form in terms of known quantities,
(7)$$\begin{array}{c} \begin{array}{c} P_{i}^{*}\left(A_{\left[0,1\right)}\times A_{1}\right)=\int_{A_{\left[0,1\right)}\times A_{1}}\frac{d\pi_{1}}{dp_{1}^{Q_{i-1}^{*}}}dQ_{i-1}^{*},\;Q_{i}^{*}\left(A_{0}\times A_{\left(0,1\right]}\right)=\int_{A_{0}\times A_{\left(0,1\right]}}\frac{d\pi_{0}}{dp_{0}^{P_{i}^{*}}}dP_{i}^{*}, \end{array} \end{array}$$
where $p_{1}^{Q_{i-1}^{*}}$ and $p_{0}^{P_{i}^{*}}$ are the marginals of $Q_{i-1}^{*}$ and $P_{i}^{*}$ at times 1 and 0, respectively. While a closed form expression to estimate half bridges is known, its components are usually not available in closed form and require approximations. Note that the time reversed formulations (Section 2.1.2) and a simpler variant of Lemma 3 also apply to the half bridge problems. Applying the disintegration theorem (see Appendix B) we can reduce the IPFP introduced to the more popular instance of IPFP proposed by [8]. Furthermore for the case of discrete measures this algorithm reduces to the Sinkhorn–Knopp algorithm [3].

## 3. Methodology

In this section we introduce IPML by providing the theoretical foundations of our algorithm and convergence guarantees. Then, we propose a practical implementation of IPML based on a Bayesian non-parametric model (GP). These components allow us to solve the general SBP problem.

### 3.1. Approximate Half Bridge Solving as Optimal Drift Estimation

We present a novel approach to approximately solve the empirical Schrödinger bridge problem by exploiting the closed form expressions of the half bridge problem. Rather than parametrising the measures in the half bridge and solving the optimisation numerically, we seek to directly approximate the measure that extremises the half bridge objective. We achieve this by using Gaussian processes [11] to estimate the drift of the trajectories sampled from the optimal half bridge measure.

We start from the following observation, which tells us how to sample from the optimal half bridge distribution (see Appendix C for proof).

**Observation** **1.**
*We can parametrise a measure Q with its drift as either of the SDEs,*

$$\begin{array}{c} \begin{array}{c} dx^{\pm }\left(t\right)=b^{\pm }\left(t\right)+\sqrt{\gamma }d\beta^{\pm }\left(t\right),\;\;x^{\pm }\left(0\right)∼\pi_{0,1}^{Q}. \end{array} \end{array}$$

*Then, we can sample from the solution to the half bridges,*

$$\begin{array}{c} \begin{array}{c} P^{*-}\;\;=\;\;\underset{P\in D(\cdot ,\pi_{1})}{arg inf}\;\;D_{\mathrm{KL}}\left(P||Q\right),\;P^{*+}\;\;=\;\;\underset{P\in D(\pi_{0},\cdot )}{arg inf}\;\;D_{\mathrm{KL}}\left(P||Q\right), \end{array} \end{array}$$

*via simulating trajectories (e.g., using the Euler–Maruyama (EM) method) following the SDEs*

$$\begin{array}{c} \begin{array}{c} dx^{\pm }\left(t\right)=b^{\pm }\left(t\right)+\sqrt{\gamma }d\beta^{\pm }\left(t\right),\;\;x^{\pm }\left(0\right)∼\pi_{0,1}. \end{array} \end{array}$$

*Solutions to the above SDEs are distributed according to $P^{*-}$ and $P^{*+}$, respectively.*


Intuitively, we are performing a cut-and-paste-styled operation by cutting the dynamics of the shortened (unconstrained) time interval and pasting the constraint to it at the corresponding boundary.

**Theorem** **1.**
*(Consistency of Reverse-MLE Formulations) Let ${\left\{{\left(x_{t_{k}}^{(n)+}\right)}_{k=0}^{T}\right\}}_{n=0}^{N}$ be sampled/discretised trajectories from the SDE that represents the half bridge measure $P^{+}$:*

$$\begin{array}{c} \begin{array}{c} dx^{+}\left(t\right)=u_{0}^{+}\left(x^{+}\left(t\right),t\right)+\sqrt{\gamma }d\beta^{+}\left(t\right),\;x\left(0\right)∼\pi_{0}. \end{array} \end{array}$$

*Then carrying out maximum likelihood estimation of the time reversed drift ${u_{0}}^{-}$ on time reversed samples:*

(8)
$$\begin{array}{c} \begin{array}{c} \prod_{n}\;p\;\left(\;{\left(x_{t_{k}}^{(n)+}\right)}_{k=0}^{T}|u^{-}\right)\;\;\propto \;\prod_{n}\;p\left(x_{t_{T}}^{(n)+}\;\right)\;\prod_{k=0}^{T}\;N\left(\;x_{t_{k}-\Delta t}^{(n)+}|x_{t_{k}}^{(n)+}\;\;\;\;-\;\Delta tu^{-}\;\left(\;x_{t_{k}}^{(n)+}\;\;,\;1\;-\;t_{k}\;\right),\;\gamma \Delta t\right)\;, \end{array} \end{array}$$

*where $u^{-}$ is the drift of our estimator SDE:*

(9)
$$\begin{array}{c} \begin{array}{c} dx^{-}\left(t\right)=u^{-}\left(x^{-}\left(t\right),t\right)+\sqrt{\gamma }d\beta^{-}\left(t\right),\;x^{-}\left(t\right)\left(0\right)∼p, \end{array} \end{array}$$

*converges in probability to the to the true dual drift $u_{0}^{-}\left(x,t\right)=u_{0}^{+}\left(x,t\right)-\gamma \nabla_{x}\ln p\left(x,t\right)$ where $u_{0}^{+}\left(x,t\right)$ is the optimal half bridge drift for $P^{+}$ as N→∞,Δt→0 (see Appendix D for proof).*


Note that w.l.o.g. the above result also holds for estimating the forward drift from backward samples. The combination of Observation 1 and Theorem 1 constitute one the main contributions of this work as they allow us to solve half bridges with a simple regression objective. Finally it is important to highlight that in practice for most interesting SDEs we can only sample approximately using schemes such as the Euler Mayurama (EM) method, for these settings our proof of Theorem 1 does not hold, however we believe it may be possible to extend/adapt the result for the approximate EM sample case.

### 3.2. On the Need for Time Reversal

We have already provided the intuition as to how time reversal enables the exchange of initial value constraints for final value constraints. In what follows we provide a more detailed technical review of this important simplification. Consider the first half bridge problem for a given IPFP iteration, $P^{*-}={arg inf}_{P\in D(\cdot ,\pi_{1})}D_{\mathrm{KL}}\left(P||Q\right)$. When formulated without time reversal in terms of the forward drift and the forward diffusion it reduces to a stochastic control problem which is subject to dynamics,
$$\begin{array}{c} \begin{array}{c} dx^{+}\left(t\right)=b^{+}\left(t\right)+\sqrt{\gamma }d\beta^{+}\left(t\right),\;x\left(1\right)∼\pi_{1}, \end{array} \end{array}$$
which is a forward SDE with a terminal hitting condition. Unlike a forward SDE with an initial value problem the above SDE is not trivial to sample from, one approach would be to solve the backwards Kolmogorov equation subject to the terminal condition; however, this approach requires (1) solving a parabolic PDE that typically involves mesh-based methods that do not scale well in high dimensions and (2) carrying out density estimation on the samples from $\pi_{1}$, also problematic in high dimensions; however, if we consider the time reversed process, the terminal condition becomes an initial value problem, $dx^{-}\left(t\right)=b^{-}\left(t\right)+\sqrt{\gamma }d\beta^{-}\left(t\right),x\left(0\right)∼\pi_{1},$ for which it is easy to sample consistent trajectories via the Euler–Maruyama (EM) discretisation without mesh-based methods or additional density estimations.

### 3.3. Iterative Proportional Maximum Likelihood (IPML)

Combining the KL minimisation routines in the original IPFP algorithm with our MLE-based drift estimation provides Figure 3. The routine DriftFit fits a dual drift on the EM sampled trajectories, and can be parametrised by any function estimation procedure with consistency guarantees. The SDESolve routine generates *M* trajectories using the EM method.

**Drift Estimation with Gaussian Processes**: One can choose any parametric or non parametric model to carry out the DriftFit routine. In this section, we briefly introduce our implementation. Our DriftFit routine is based on the work in [12,23] that uses GPs to estimate the drift of SDEs from observations. We refer to this routine as GPDriftFit. We can restate the regression problems given by Equation (Equation 8) and its reversed counterpart in the following form:$$\begin{array}{cc} \frac{x_{t_{k}}^{(n)-}\;\;\;\;\;-\;x_{t_{k}-\Delta t}^{(n)-}}{\Delta t}\; & =\;u^{+}\;\left(\;x_{\left(t_{k}\;-\Delta t\right)}^{(n)-},1-\left(t_{k}\;-\;\Delta t\right)\right)\;+\;\frac{\sqrt{\gamma }}{\Delta t}\epsilon ,\\ \frac{x_{t_{l}\;\;-\Delta t}^{(m)+}\;\;-x_{t_{l}}^{(m)+\;\;}}{\Delta t}\; & =\;-u^{-}\left(\;x_{t_{l}}^{(m)+},1-t_{l}\right)\;+\;\frac{\sqrt{\gamma }}{\Delta t}\epsilon , \end{array}$$
where $\epsilon ∼N(0,I_{d})$. Placing GP priors on the drift functions $u^{-}∼\mathrm{GP}$ and $u^{+}∼\mathrm{GP}$, we arrive at standard multi-output GP formulation. Following [12,24], we assume the dimensions of the drift function are independent (assuming that the drift dimensions are decoupled is the least restrictive assumption as coupling them would impose a form of regularization). (equivalent to imposing a block-diagonal kernel matrix on a multi-output GP) and thus we fit a separate GP for each dimension. See Section E.1 for the specification of the predictive means.

Similar to [12,23], we are not making use of the predictive variances (using the predictive mean as an estimate for the drift can also be interpreted as a form of kernel ridge regression under the empirical risk minimisation framework). Instead, we simply use the predictive mean as an estimate of the drift and subsequently use that estimate to perform the EM method, thus effectively we could interpret this approach as a form of kernel ridge regression under the empirical risk minimisation framework.

Note that the advantage of the Bayesian GP interpretation of this procedure is that the GP posterior under certain conditions is naturally conjugate to the posterior of the SDE drift when modelled with a GP prior [23,25]. Furthermore, the Bayesian non-parametric formulation allows for the encoding of the prior drift function as the mean function of the GP prior (i.e., $u\;∼\;GP(b_{0}^{Q_{0}^{\gamma }},k)$). This becomes helpful when trying to sample unlikely paths and evaluating the fitted drift at samples that were not observed by the GP during training, here we want the fitted drift to fall back to the prior rather than to a Brownian motion given by a GP prior with 0 mean.

We now have the relevant ingredients to carry out IPML as specified in Figure 3. The computational cost of IPML with a Gaussian process is detailed in Section H.5. The majority of the computation is spent fitting the GP in DriftFit at each iteration and scales as a function of the time discretization used as well as the size of $\pi_{0}$ and $\pi_{1}$.

## 4. Related Methodology

In this section we carry out a conceptual comparison with two pre-existing numerical approaches for solving the static SBP. While our goal is to solve the dynamic SBP, the solution of the static SBP can be used to construct that of the dynamic SBP [10], and this connection is central to our discussion. We would also like to highlight that an algorithm akin to IPML has been proposed concurrently and independently by [26], the main difference with our algorithm is that they estimate the drifts of the SDEs using neural networks score matching while we use using Gaussian processes and maximum-likelihood-based ideas.

### 4.1. Sinkhorn–Knop Algorithms

Within the machine learning community the static SBP with a Brownian motion prior is popularly known as entropic optimal transport [4]. In this formulation there are no trajectories and the empirical distributions are treated as discrete measures ${\left[\pi_{0}\right]}_{j}\;\;=\;\frac{1}{N}$ and ${\left[\pi_{1}\right]}_{i}\;\;=\;\frac{1}{M}$ thus the objective is given by $\min_{Q\in D(\pi_{0},\pi_{1})}\langle Q,C^{Q_{0}^{\gamma }}\rangle +\gamma h\left(Q\right)$, where *h* is the discrete entropy, 〈·,·〉 computes the dot product between two matrices. The cost matrix *C* corresponds to the log transition density induced by the prior SDE: $C_{ij}^{Q_{0}^{\gamma }}=\ln p^{Q_{0}^{\gamma }}\left(y_{i}|x_{j}\right)$, which in the case of a Brownian motion prior reduces to a scaled Euclidean distance.

Once the problem has been discretized as described, the Sinkhorn–Knopp algorithm [3] can be applied directly to fit an optimal discrete transport map (and discrete SBP potentials $\phi ,\hat{\phi }$) between the two distributions. Recent work [6] has showed empirical success in forming a continuous approximation of the SBP potentials using the logsumexp formula [6]; however, it still remains to formally analyse the accuracy of the logsumexp potentials.

Estimating the cost matrix $C_{ij}^{Q_{0}^{\gamma }}$ required by the Sinkhorn algorithm requires a mixture of both density estimation and further simulation of the prior. Furthermore, once we have the logsumexp potentials, additional high dimensional integrals must be estimated every time we wish to evaluate the optimal drift. Details are discussed in Appendix F. In addition, the Sinkhorn–Knopp algorithm still faces challenges in high dimensional spaces, specially for small values of γ [27]. Many proposed enhancements and literature [4,5] have focused on cost functions that implicitly require a Brownian motion prior and thus do not apply to our general setting.

### 4.2. Data-Driven Schrödinger Bridge (DDSB)

The method proposed by [10] is perhaps the most similar approach to our approach and consists of iterating two coupled density estimation objectives (see Appendix G) fitted at the marginals until convergence. While conceptually similar to our approach, there are three key differences. As with Sinkhorn–Knopp-based methods, their approach aims to solve the static SBP. Once converged, it requires further approximation to estimate the optimal drift. The coupled maximum likelihood formulation of the static half bridges in [10] is based on un-normalised density estimation with respect to the SBP potentials $\phi ,\hat{\phi }$. The coupling of $\phi ,\hat{\phi }$ in these objectives does not directly admit the application of modern methods in density estimation since it does not allow us to freely parametrise MLE estimators for the boundaries, and thus, neural density estimators such as [28,29,30] cannot be taken advantage of to circumvent the computation of the partition function. Regression problems are ubiquitous in machine learning (ML) and thus why we believe that this formulation can be very impactful as it allows us to leverage all these methods from ML. Experimentally, regression methods have been observed to scale better to high dimensional problems than density estimation methods, we observed similar evidence as we were unable to scale up the DDSB method beyond two dimensions.

Additionally, to compute the normalising term in the DDSB objective, we have to estimate a multidimensional integral that is not taken with respect to a probability distribution. This poses a difficult challenge in high dimensions. For a more detailed commentary, see Appendix G. Note that the method by [20] uses importance sampling to estimate these quantities which performs poorly beyond two dimensions.

It worth noting that we are interested in a method that can obtain a dynamic interpolation between two distributions. A downside of solving the static bridge either by Sinkhorn or [10] is that it does not directly provide us with an estimate of the optimal dynamics. In order to obtain an estimate of the optimal drift we require a series of approximations to estimate the integrals in Equations (Equation 88) and (Equation 89), thus every time we evaluate the optimal drift using these approaches we have to simulate the prior SDE O(N+M) times. This makes the runtime of obtaining the drift and dynamical interpolation expensive, since for each Euler step we take we have to simulate another SDE and backpropagate through it to evaluate the drift.

Finally we would like to highlight that a series of modern approaches to generative modelling [26,31,32,33] motivate both empirically and theoretically the gain in accuracy obtained in generative modelling tasks when using a dynamical approach rather than a static one.

## 5. Numerical Experiments

In this section, we demonstrate the capability of IPML to solve the Schrödinger bridge while efficiently incorporating priors on a range of different tasks from synthetic experiments to embryo cells (code supporting experiments can be found at https://github.com/franciscovargas/GP_Sinkhorn, accessed on 22 August 2021).

### 5.1. Simple 1D and 2D Distribution Alignment

The first experiment considered is a simple alignment experiment where $\pi_{0}$ and $\pi_{1}$ are either unimodal or bimodal Gaussian distributions (see Section H.1 for exact details on the distributions). In Table 1 we compare the accuracy of the fitted marginals with our implementation of the Data-Driven Schrödinger Bridge (DDSB) by [10]. The scoring metrics used are the Earth mover’s distance (EMD) as well as a Kolmogorov–Smirnov (KS) statistic on both sample sets.

We were unable to get DDSB to work well when $\pi_{0}$ and $\pi_{1}$ were distant from each other. In these distant settings, DDSB collapses the mass of the marginals to a single data point (see Section H.1.1), as a result for the unimodal experiment we had to set γ=100 for the DDSB approach to yield sensible results. We can observe IPML obtains better marginals overall and at a lower value of γ=1.

We carried out 2D experiments with our approach to show the ability of our method to diffuse from a simple uni-modal distribution to a multi-modal distribution as in Figure 1 where our learned bridge successfully splits. Furthermore, we can visually observe how the forward and backwards trajectories are mirror images of each other as expected.

### 5.2. 2D Double Well Experiments

In the double well experiment, we illustrate how to incorporate an arbitrary functional prior and learn the distribution over paths connecting $\pi_{0}$ and $\pi_{1}$. In order to encode prior information, we experiment with the potential well illustrated in Figure 1 and Figure A3. The boundary distributions $\pi_{1},\pi_{0}$ are taken to be Gaussian distributions centred at the centre of each well, respectively (See Section H.2 for the experiment specification and IPML parameters).

The motivation behind this experiment is to show that the SBP with this prior follows low energy (according to the well’s potential function) trajectories for configurations of particles sampled at the wells. Intuitively, we can expect the learned trajectories to avoid the high energy peak located at x=(0,0) and go via the “passes” on either side. Note that if we estimated the optimal transport (OT) geodesics between $\pi_{0}$ and $\pi_{1}$ or similarly ran IPML with a Brownian motion prior, the learned optimal trajectories would go right through the middle, which is the highest energy path between both wells.

The prior is incorporated in the algorithm in two different ways, (1) by having the first drift $Q_{0}^{*}$ to follow the negative derivative of the potential function $dx\left(t\right)=-\nabla_{x}U\left(x\right)+\gamma d\beta \left(t\right)$ and (2) by setting the mean function of the GP used to fit the drift as the negative derivative of the potential function. As illustrated in Figure 1, the learned trajectories between the two wells avoid the main high energy region and go through via lower energy passes, thus respecting the potential prior; however, the behaviour of the trajectory will differ depending on the choice of kernel, as illustrated in Figure A3, underlying the need for its careful consideration. We compared our approach to DDSB using the mean squared error distance to the prior in our evaluation. The results were **DDSB: 22.1, IPML (no prior): 19.9, IPML: (prior) 18.7**. As we can see, IPML considerably outperforms DDSB.

### 5.3. Finite Sample/Iteration Convergence

Theorem 1 provides us with asymptotic guarantees; however, it does not extend to the finite sample and discretisation case. To highlight the importance of finite effects on IPML, we carried out this analysis empirically. In Figure 3 we plot an empirical estimate of the error term in the control formulation of the SB (Equation (Equation 5)). This term is effectively the mean squared error between the learned drift and the prior drift (gradient field of the well). We analyse this metric for different values of *N* (number of samples) and $\Delta_{t}$ (discretization factor). We observe that IPML quickly reaches a low error valley, then, the cumulative error from the successive finite sample MLE can be observed and the drift starts to slowly deviate from the prior, this motivates early stopping. As *N* is increased, IPML achieves lower error faster and deviates less from the prior in later iterations. Finally, we observe placing the drift prior via the GP has a significant effect in improving the error and its convergence. Additionally in Appendix E we detail how this question could be approached from a theoretical perspective while underlining its significance and difficulty as illustrated by the lack of such analyses in related algorithms [10,22].

### 5.4. Single Cell—Embryo Body (EB) Data Set

We perform an experiment on an embryoid body scRNA-seq time course [34]. Single-cell RNA sequencing enables accurate identification of cells at specific time points; however, all cells are destroyed by measurement. This prevents modelling single-cell trajectory and instead we rely on modelling the data-manifold at discrete time points. The datasets consist of 5 time points illustrated in Figure A4. To evaluate the performance of the algorithms, we fit the models at the endpoints (T=1,5) and predict the intermediate frames. The metric used is the Earth mover’s distance between the data at intermediate frames and the predicted distribution. We evaluate the performance at the endpoints by considering the prediction of the forward model at T=5 and the backward model at T=1.

We compare the performance with two methods. The first one, TrajectoryNet [34], uses continuous normalising flows with a soft constraint based on optimal transport. The second leverages the McCann interpolant [35,36] to interpolate the discrete OT solution; this corresponds to the linear interpolation induced by the transport map.

The results are summarised in Table 2. In most frames, IPML outperforms TrajectoryNet and performs similarly to OT. As the noise (volatility) in the trajectories goes to 0, IPML theoretically converges to the OT solution with linear geodesics. So without any additional prior information, we would not expect IPML to outperform OT; however, when dealing with finite data, the DriftFit procedure allows for some non-linearity in the trajectories if it improves the fit. As a result, we do see some differences in the convex hull displayed in Figure 4 where we observe a better coverage of the single cell observations. We hypothesize this explains the improvement in performance vs. OT for frame 4. The performance of IPML may be further improved by incorporating domain-specific knowledge as a prior. It would outperform OT in those settings.

### 5.5. Motion Capture

In this experiment, we demonstrate how IPML can be used to model human motion from sensor data (data from The CMU Graphics Lab Motion Capture Database funded by the NSF (http://mocap.cs.cmu.edu, accessed on 22 August 2021)). The motion corresponds to a basketball movement where the subject raises both arms simultaneously as illustrated in Figure 5 and each sensor corresponds to an oriented angle. We focus on modelling the right shoulder and elbow where the starting distribution corresponds to the leftmost image and the ending one the rightmost. This results in a 4-dimensional space as we model both position and velocity for each sensor. We compare the fit of IPML using a Brownian and a 2nd order linear ODE (Langevin) prior. The experimental details can be found in Section H.3. As illustrated in Figure 5, IPML is able to approximately model the dynamics using both priors. The Brownian prior displays noisy trajectories as expected in contrast to the Langevin prior that, by construction, smooths out the predicted positions (due to the 2nd order term). We observe that the Langevin prior approximates the step function nature of the true trajectory more closely than the Brownian prior, additionally, we can see that it also produces slightly better alignments.

## 6. Limitations and Opportunities

In this work, we propose to use GPs to estimate the drift; however, IPML could also be used with a parametric function estimator (e.g., a neural network). This could be useful with high-dimensional data where GPs may underperform. The main advantage of using a GP is its capacity to incorporate functional priors via the mean function. This can be useful in applications such as molecular dynamics where a potential function may be available. In contrast, implementing functional regularization in a neural network would require approximating a non-trivial high-dimensional integral to estimate the mean squared error between the parametric estimator and the functional prior.

A particular useful extension would be to adapt the SBP to work with more general forms of volatility functions that are not constant. This can be used, for example, in enforcing positivity constraints on a stochastic process via a geometric Brownian motion prior; this has applications in modelling biological signals such as transcription factors [37]. Work in this direction would require extending the theory of SBPs to non constant volatility functions.

Another promising setting is when multiple frames of data are available rather than just two boundary conditions. The IPFP algorithm trivially adapts to multiple constraints [21] rather than just initial and terminal distributions. Future work could explore experiments similar to the one presented in [34] where multiple frames are considered during training and the performance is measured using a leave-one-out procedure.

## 7. Conclusions

We have presented IPML, a method to solve the Schrödinger bridge for arbitrary diffusion priors. We presented theoretical results guaranteeing convergence in the limit of infinite data. We devised a practical application of the algorithm using Gaussian processes and presented several experiments on a variety of problems from synthetic to biological data. The approach opens up opportunities in science, where oftentimes, prior knowledge about the temporal evolution of a process has been developed but needs to be combined with data-driven methods to scale up to modern problems.

## Figures and Tables

**Figure 1 entropy-23-01134-f001:**
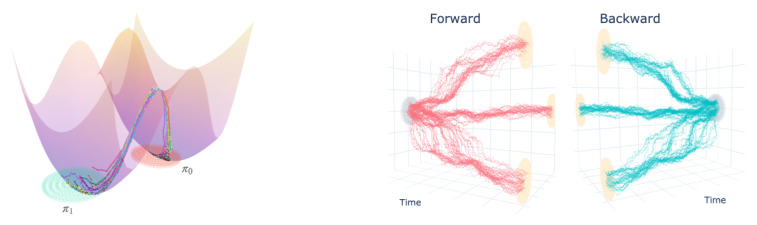
**Left**: Learned SBP trajectories in the double well experiment of Section 5.1, with prior $Q_{0}^{\gamma }$ expressed in terms of an energy landscape U(x,y) as $dx\left(t\right)=-\nabla_{x}U\left(x\left(t\right)\right)+\gamma d\beta \left(t\right)$. **Right:** Forwards and backwards diffusion of learned SBP between unimodal and multimodal boundary distributions (see Section 5 for experimental details).

**Figure 2 entropy-23-01134-f002:**
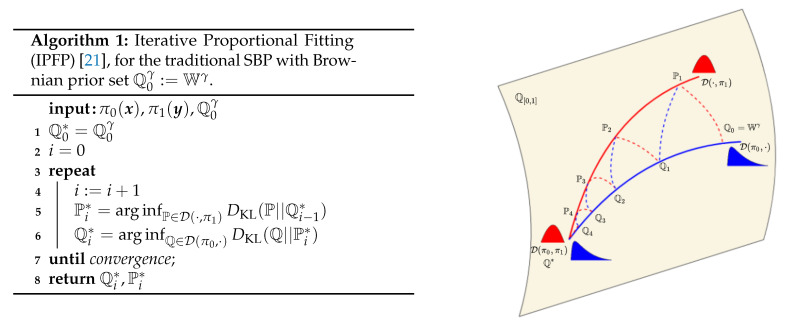
**Left**: IPFP algorithm. **Right**: illustration of the convergence of the IPFP Algorithm.

**Figure 3 entropy-23-01134-f003:**
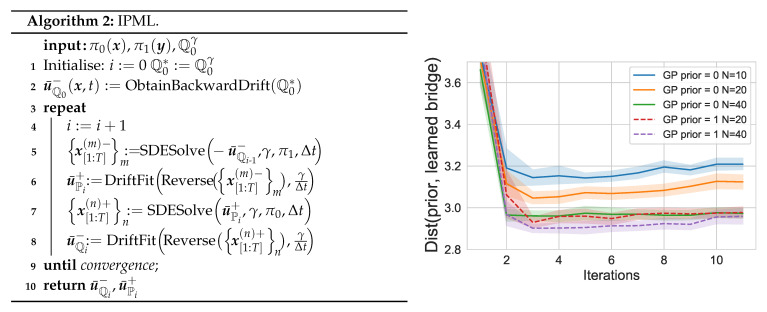
Distance between the prior and the learned bridge as a function of the iterations, number of samples used (N) and whether the prior is used for the GP ($T\;=\;\frac{1}{\Delta t}\;=\;100$).

**Figure 4 entropy-23-01134-f004:**
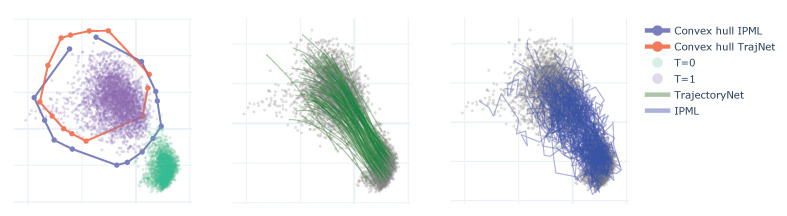
The left plot represent the first and last frame of the cell’s data and the convex hull of the forward model for TrajectoryNet and IPML. The right plots represent sample trajectories for Trajectorynet and IPML.

**Figure 5 entropy-23-01134-f005:**
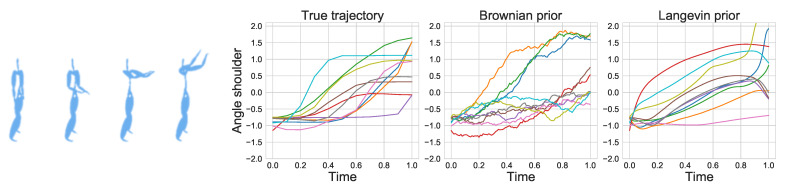
**Left**: 3D animation of the basketball signal motion modeled in Section 5.4 (motion from left to right). **Right**: Trajectories of the shoulder’s oriented angle sensor through the motion. The two plots on the right demonstrate IPML’s fit using a Brownian and Langevin prior.

**Table 1 entropy-23-01134-t001:** Performance comparison of the fitted marginals between DDSB and IPML on unimodal and bimodal experiment.

	Unimodal				Bimodal			
	$\pi_{0}$		$\pi_{1}$		$\pi_{0}$		$\pi_{1}$	
	KS	EMD	KS	EMD	KS	EMD	KS	EMD
DDSB	0.17	0.34	0.19	0.13	0.18	0.12	0.07	0.04
IPML	0.06	0.10	0.13	0.04	0.05	0.04	0.07	0.15

**Table 2 entropy-23-01134-t002:** Earth Moving Distance (EMD) on the EB data. EXP stands for the exponential kernel and EQ for exponentiated quadratic. The column “full” represents EMD averaged over all frames whereas “path” is averaged over the intermediate frames (T=2,3,4).

	T = 1	T = 2	T = 3	T = 4	T = 5	Mean	
						Path	Full
TrajectoryNet	0.62	1.15	1.49	1.26	0.99	1.30	1.18
IPML EQ	0.38	1.19	1.44	1.04	0.48	1.22	1.02
IPML EXP	0.34	1.13	1.35	1.01	0.49	1.16	0.97
OT	Na	1.13	1.10	1.11	Na	1.11	Na

## Data Availability

Publicly available datasets were analysed in this study. These datasets can be found here: Motion Capture Dataset: http://mocap.cs.cmu.edu. Mission statement for dataset: http://mocap.cs.cmu.edu/info.php. Instructions to download EB single cell dataset can be found at https://nbviewer.jupyter.org/github/KrishnaswamyLab/PHATE/blob/master/Python/tutorial/EmbryoidBody.ipynb. The dataset itself can be found at https://data.mendeley.com/datasets/v6n743h5ng/1. All websites were accessed on 22 August 2021.

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
