# Peer review of "Solving Schrödinger Bridges via Maximum Likelihood"

_entropy, 2021, doi:10.3390/e23091134_

Round 1

Reviewer 1 Report

This is a fine, interesting article. I have, however, a few concerns:

1) Originality: the totality (or nearly so) of theorems and lemmas presented are known, as are the main ideas of the algorithm proposed. Using the reverse process is key to the Sinkhorn iteration, which dates back to Schroedinger himself and has been used in nearly all later work. The use of half bridges and of maximal likelihood has been proposed in reference [18] (which has since been published in CPAM.) The authors should make clearer what the original contribution of this article is (the parameterization of the diffusion through Gaussian processes?)

2) What the authors call "the static problem", i.e. the coupling between the initial and final distributions, can be solved without reference to the diffusion process in between, except for the prior it generates on this coupling. Instead, the methodology proposed iterates on the the diffusion process itself. This is presented as an advantage of the methodology, but at first sight it appears as a drawback, as estimations need to be made in the much larger space of diffusion processes, which include not just the two marginals but the set of all paths in between. The authors should clarify in what sense this is advantageous.

3) The comparison with reference [18], with many commonalities with the methodology proposed, appears somewhat peculiar.  For instance, the sentence

  1. The coupling of φ, φˆ in these objective does not directly admit the application of

  2. 274  modern methods in density estimation.

is empty of content. And then,

"

  1. We were unable to get DDSB to work well when π0 and π1 were distant from each

  2. 290  other. In these distant settings, DDSB collapses the mass of the marginals to a single

  3. 291  data point (see Appendix H.1.1), as a result for the unimodal experiment we had to set

  4. 292  γ = 100 for the DDSB approach to yield sensible results. We can observe IPML obtains

  5. 293  better marginals overall and at a lower value of γ = 1."

seems more a reflection on the authors' limitations and lack of consultation with the authors of [18] than a commentary on that work.

This connects to the previous two points, and should also be clarified.

4) The assumptions that the drift decouples by dimension and that the variance of the diffusion process is uniform appear as severe limitations, which contradict the initial statement on the generality of diffusion processes considered. This also needs clarification.

5) A very minor point: the current abstract includes the instructions for preparing the abstract! Please correct.

Author Response

1. Originality: the totality (or nearly so) of theorems and lemmas presented are known ...

The main novel contribution of this work is the reformulation of the dynamic version of the Sinkhorn algorithm (Dynamic IPFP) to a maximum likelihood estimation (MLE)  problem, but more specifically a regression problem. This is formalised via Observation 1 and Theorem 1 which to the best of our knowledge are both novel results which we prove in the Appendix. This is different to the DDSB approach by Pavon et al. in which the MLE characterisation is a density estimation problem. This contribution is what allows us to parametrize an approximation to the dynamic Sinkhorn iterates with GPs that allow incorporating the SBP prior into the drift, this is the secondary  contribution of this work. It is important to note that instead of a GP one could use a neural network to solve the regression problem, this is something that we have experimented with and some results can be found in the public code repository for this paper. Regression problems are ubiquitous in machine learning (ML) and thus why we believe that this formulation can be very impactful as it allows us to leverage all these methods from ML. Experimentally, regression methods have been observed to scale better to high dimensional problems than density estimation methods. We also observed this issue in our experiments where scaling the density-estimation based methods from [Pavon et al. 2018] above 3 dimensions becomes challenging. We have added some bullet points at the end of the Introduction to emphasize the novel contributions.

2. What the authors call "the static problem", i.e. the coupling between the initial and final ...

This is an excellent point and we have added a short paragraph at the end of the methods section to highlight some of the challenges posed by the static SBP methods.

It is important to highlight that we are interested in a method that can obtain a dynamic interpolation between two distributions. A downside of solving the static bridge either by Sinkhorn or [Pavon et al. 2018] is that it does not directly provide us with an estimate of the optimal dynamics. For example, in molecular dynamics, it may be useful to examine the obtained dynamics to analyze whether they violate known physical laws. 

 As the reviewer points out the dynamic SBP brings in an added complexity at training time in that now we are considering an estimation problem that scales with the number of timesteps. However, the focus in our work is to obtain a dynamic interpolation between two datasets, in order to do this one must eventually go back to the dynamical formulation of the problem. Approaches that solve the static bridge can then be used to recover the drift of the dynamic SBP. However, this would require a Monte Carlo approximation that involves running the prior SDE for each data point, this must be done every time we evaluate the drift. Whilst our approach requires simulating dynamics at training time, once the IPFP iterations are done we are left with a drift that we can tractably evaluate without any further Monte Carlo estimation, meanwhile the static bridge approaches require simulating the dynamics at test time for each drift evaluation  if we wish to interpolate between the two datasets. To summarise this point when one has to recover the dynamics from the static bridge the problem becomes similarly complex as the dynamic formulation.

Another important point to highlight is that if our prior SDE in the SBP does not have a closed form transition (i.e. the SDE is not linear) then static methods such as DDSB will still require to run a simulation of the SDE with Euler's method in order to sample from the transition density at the end points which is required to estimate the static half bridges see bullet point 3 at end of page 23 and start of page 24 of [Pavon et al. 2018]. In summary even in the static bridge for the case of more general prior SDEs the dynamics must be simulated at each step, in short, in the general setting it is difficult to avoid the dynamical component.

Finally recent work on generative modelling in the machine learning community [Song 2021 et al, De Bortoli et al. 2021, Kingma et al. 2021] has empirically highlighted how the diffusion component seems to give higher quality samples at the marginals/boundaries, we empirically observe the same result in our experiments in Section 1. 

We have added the aforementioned discussion points to the end of section 3 in the revised manuscript.

3. The comparison with reference [18], with many commonalities with the methodology proposed ... is empty of content ...

We thank the reviewer for pointing out the missing explanation in this sentence, and completely agree that in its current form it is empty of content. We have modified it to:

The coupling of φˆ,φ in these objective does not directly admit the application of modern methods in density estimation, since it does not allow us to freely parametrise the MLE estimators for the boundaries thus neural density estimators such as [28–30] cannot be taken advantage of and will not be able to circumvent the computation of the partition function.

To expand on this, we are referring to modern neural density estimation approaches that circumvent the estimation of a partition function using a change of variables which provide the normalisation of the density via a jacobian determinant that can be computed via backpropagation circumventing any sampling. Unfortunately, the coupling of the potentials does not allow to exploit this change of variables trick and as a result require the estimation of a multidimensional integral. 

> ... seems more a reflection on the authors' limitations and lack of consultation with the authors  ...

Throughout implementing and debugging the DDSB approach by Pavon et al. we reached to authors Michelle Pavon and Giulio Trigila. We exchanged several emails throughout this process and shared some of these plots/results. It was very helpful to contact and collaborate with the authors as it clarified many aspects of their approach, as part of this collaboration process we wrote the following document https://hackmd.io/@eCr93iNpRCeZ5wUIIjiYug/SkCd41_OO in which you can find a succinct HTML-Latex  table where each line in our code is aligned to the corresponding mathematical equation from the DDSB method. This table was shared with Dr Trigila who confirmed that the implementation and understanding of their approach seemed to be accurate. In short, we did contact and collaborate with the authors (who were unable to provide any code) and dedicated a thorough effort in implementing, understanding and ensuring the correctness of their work. 

4. The assumptions that the drift decouples by dimension and that the variance of the diffusion process is uniform appear as severe limitations ..

By uniform variance do you mean that the diffusion coeficient is time homogeneous (that is it does not depend on time) ? If that is the case we never claim that the diffusion coefficient (the variance/volatility) is non-uniform (we actually define it to be uniform from the beginning). Furthermore all previous work and literature on the Schrodinger bridge problem [Schrodinger 1931, Pavon et al. 2018, Leonard , 2013, Brenton et al. 2019, De Bortoli et al. 2021] also have a uniform variance as this is how the Schrodinger bridge problem is traditionally defined. Furthermore the variance is not part of the optimisation process as it must remain equal to the prior’s variance in order for the KL divergence to remain finite (otherwise it blows up). We state in the definition that the dynamic bridge’s volatility  $\gamma$ (the diffusion coefficient/variance) is a constant and thus uniform (time-homogenous) and thus we never claim the diffusions we work with to be more general than this.  We have added a note in lines 75-77 to remind/clarify this.

> The assumptions that the drift decouples by dimension ..

The uncorrelated assumption we make among drift dimensions is equivalent to having a different function estimator per drift and is the least restrictive parameterization. Note that each drift dimension $b_i$ depends on all the dimensions of the SDE $b_i = f_i(x_1, … , x_d)$ and thus we are making no assumption regarding independence of the input dimensions. From a function estimation viewpoint purely this does not imply any strong limitations, in fact the parameterization of the drift is as expressive as if we correlated the dimensions. What might help clarify this point is if we think about our estimation in the frequentist framework. The predictive mean of a GP is in fact equivalent to minimising a regularized empirical risk over a reproducing kernel Hilbert space (RKHS) ; this is known as kernel ridge regression. Parametrizing each drift component with a different function living in the same RKHS (i.e.  $b_i = f_i(x_1, … , x_d)$ ) does not introduce any limitations from the function estimation viewpoint, that is how far are we from the true drift. This is in fact analogous to having different parameters $b_i = f(x_1, … , x_d; \theta_i)$  for each dimension when using a parametric estimator, as long as the function class  of $f$ is flexible (e.g. a neural network) enough then this is not a limitation from the viewpoint of estimating $\mathbf{b}$ it's just that we consider estimating each of its components separately (with a different function estimator).

Finally it is worth noting that correlating the GP outputs (i.e. coupling the drift dimensions) is more restrictive than the decoupled approach we have taken (which is more flexible and thus more data dependent). This point is discussed and motivated in [Alvarez et al. 2012, Evgeniou  et al. 2005], where it is illustrated that the coupling of the drift outputs correspond to regularising the RKHS hypothesis space. So in fact the decoupled approach we take in this work is more flexible (less limited) than coupling the drift outputs as discussed in [Alvarez et al. 2012, Evgeniou  et al. 2005]. It is however important to note that the regularisation effects of coupling may be desirable in certain physical systems where we wish to impose a constraint, however this would come at a large computational cost.  We have added a paragraph detailing the above to appendix E subsubsection E.1.1. 

[Schrodinger 1931] Erwin Schr¨odinger. Uber die umkehrung der naturgesetze ¨ . Verlag Akademie der wissenschaften in kommission bei Walter de Gruyter u. Company, 1931. 

[Pavon et al. 2018] Pavon, M., Trigila, G. and Tabak, E.G., 2021. The Data‐Driven Schrödinger Bridge. Communications on Pure and Applied Mathematics, 74(7), pp.1545-1573.

[Leonard , 2013] Léonard, C., 2013. A survey of the schr\" odinger problem and some of its connections with optimal transport. arXiv preprint arXiv:1308.0215.

[Song et al. 2021] Song, Y., Sohl-Dickstein, J., Kingma, D.P., Kumar, A., Ermon, S. and Poole, B., 2020. Score-based generative modeling through stochastic differential equations. arXiv preprint arXiv:2011.13456.

[De Bortoli et al. 2021] De Bortoli, V., Thornton, J., Heng, J. and Doucet, A., 2021. Diffusion Schr\" odinger Bridge with Applications to Score-Based Generative Modeling. arXiv preprint arXiv:2106.01357.

[Kingma et al. 2021] Kingma, D.P., Salimans, T., Poole, B. and Ho, J., 2021. Variational Diffusion Models. arXiv preprint arXiv:2107.00630.

[Brenton et al. 2019] Bernton, E., Heng, J., Doucet, A. and Jacob, P.E., 2019. Schr\" odinger Bridge Samplers. arXiv preprint arXiv:1912.13170.

[Huang et al. 2021] Huang, J., Jiao, Y., Kang, L., Liao, X., Liu, J. and Liu, Y., 2021. Schr {\" o} dinger-F {\" o} llmer Sampler: Sampling without Ergodicity. arXiv preprint arXiv:2106.10880.

[Alvarez et al. , 2012] Álvarez, M.A., Rosasco, L. and Lawrence, N.D., 2012. Kernels for Vector-Valued Functions: A Review. Foundations and Trends® in Machine Learning, 4(3), pp.195-266.

[Evgeniou  et al. 2005] Evgeniou, T., Micchelli, C.A., Pontil, M. and Shawe-Taylor, J., 2005. Learning multiple tasks with kernel methods. Journal of machine learning research, 6(4).

Reviewer 2 Report

I have analyzed the manuscript, it has interesting results which can be useful for the researchers on this subject. I recommend for publication this manuscript after the authors check the manuscript in order to avoid, as present in the abstract, the texts as follows:

"A single paragraph of about 200 words maximum. For research articles, abstracts should give a pertinent overview of the work. We strongly encourage authors to use the following style of structured abstracts, but without headings: (1) Background: place the question addressed in a broad context and highlight the purpose of the study; (2) Methods: describe briefly the main methods or treatments applied; (3) Results: summarize the article’s main findings; (4) Conclusion: indicate the main conclusions or interpretations. The abstract should be an objective representation of the article, it must not contain results which are not presented and substantiated in the main text and should not exaggerate the main conclusions."

Author Response

We would like to thank the reviewer for taking the time to read our work and the kind comments. We have fixed the mistake in the abstract in our revised version as well as clarified the contributions of this work more precisely across the manuscript.

Round 2

Reviewer 1 Report

The authors have addressed all the concerns that I had raised, I believe that the article is suitable for publication.